# Renal Cell Carcinoma in End-Stage Kidney Disease and the Role of Transplantation

**DOI:** 10.3390/cancers16010003

**Published:** 2023-12-19

**Authors:** Samuel Robinson, Alena Nag, Benjamin Peticca, Tomas Prudencio, Antonio Di Carlo, Sunil Karhadkar

**Affiliations:** 1Lewis Katz School of Medicine, Temple University, Philadelphia, PA 19140, USA; samuel.robinson0002@temple.edu (S.R.); ben.peticca@temple.edu (B.P.); tuo35905@temple.edu (T.P.); adc@temple.edu (A.D.C.); 2Department of Surgery, Temple University Hospital, Philadelphia, PA 19140, USA; alena.nag@temple.edu

**Keywords:** renal cell carcinoma, end-stage renal disease, transplantation, immunosuppression, renal malignancy, kidney transplant, dialysis

## Abstract

**Simple Summary:**

This review aims to characterize the unique presentation of renal cell carcinoma (RCC) in kidney transplant patients with end-stage kidney disease (ESKD). RCC in the ESKD patient is an ambiguous and under-studied area of oncology, as dialysis and transplantation alter the biological environment of the kidney. Assessing outcomes and dictating management for RCC in this anomalous patient population pose oncologic challenges, and thus warrant a review to update clinicians. The potential for donor tissue to cause cancer in the organ recipient is unknown, but one can assume that this risk is minimal. Our review challenges this concept and we propose new insights into how donor and recipient environments might influence the risk of cancer in the transplant recipient.

**Abstract:**

Kidney transplant patients have a higher risk of renal cell carcinoma (RCC) compared to non-transplanted end-stage kidney disease (ESKD) patients. This increased risk has largely been associated with the use of immunosuppression; however, recent genetic research highlights the significance of tissue specificity in cancer driver genes. The implication of tissue specificity becomes more obscure when addressing transplant patients, as two distinct metabolic environments are present within one individual. The oncogenic potential of donor renal tissue is largely unknown but assumed to pose minimal risk to the kidney transplant recipient (KTR). Our review challenges this notion by examining how donor and recipient microenvironments impact a transplant recipient’s associated risk of renal cell carcinoma. In doing so, we attempt to encapsulate how ESKD-RCC and KTR-RCC differ in their incidence, pathogenesis, outcome, and approach to management.

## 1. Introduction

It is estimated that as many as 9.7 million people suffer from end-stage kidney disease (ESKD) worldwide [1]. Renal replacement therapy (RRT), clinically defined as dialysis with or without renal transplantation, remains the mainstay treatment modality for ESKD. Compared to the general population, ESKD patients on long-term dialysis are at a greater risk of developing renal cell carcinoma (RCC) [2]. These patients go on to assume additional risk temporarily after receiving a life-saving kidney transplant. Dialysis and transplantation are known to alter the biologic environment in renal tissue, increasing the potential for tumor growth through different mechanisms.

It has long been suggested that RCC in an ESKD patient may behave as a distinct biological entity compared to RCC in a non-ESKD patient. This notion is supported by the epidemiology and pathophysiology of RCC in ESKD renal tissue. However, this is further complicated by the bidirectional connection between ESKD and RCC [3]. Treatment modalities for RCC, including both nephrectomy and anti-cancer medication, reduce functional renal tissue, thus increasing one’s risk of CKD and ESKD. Similarly, ESKD may lead to acquired cystic disease (ACD) and increased reactive oxidative species, both of which increase one’s risk of RCC [4]. Transplantation in these patients adds to the complex bidirectional cause and effect relationship seen between RCC and ESKD, as shown in Figure 1.

Kidney transplantation introduces foreign renal tissue carrying its own oncogenic potential unrelated to the recipient, who bears their own genetic predisposition to cancer. Beyond this, post-transplant immunosuppression is hypothesized to produce a pro-oncogenic environment in the kidney, which may allow for unregulated cell growth and proliferation. Many argue a similar physiological mechanism can explain the increased prevalence of RCC in ESKD patients [5]. ESKD-associated uremia may dampen a patient’s immune system, supporting cancerous tumor growth in the kidney. Clinicians managing ESKD patients must balance the associated risk of kidney failure with the risk of neoplastic growth due to post-transplant immunosuppression therapy and the addition of novel donor tissue.

With RCC accounting for over 2% of all cancer deaths in 2021 [6], it is vital to better understand the patient population who is at the greatest risk: ESKD patients undergoing kidney transplantation. This review attempts to characterize the difference between ESKD-RCC and post-transplantation RCC to illuminate the manner in which immunosuppression and donor renal tissue impact the risk of RCC and the subsequent oncologic outcomes.

## 2. Incidence and Risk Factors

In 2022, cancer of the kidney accounted for 4.1% of all new cancer cases and 2.3% of all cancer deaths [7]. RCC represents 94% of all kidney cancer, with cancer of the renal pelvis and Wilms tumors making up the remaining 6% [8]. From 2008 to 2017, the incidence of kidney cancer has risen by 1% each year [8]. In 2015, as many as 9.7 million people were estimated to suffer from ESKD [1]. Risk factors associated with RCC include older age, smoking history, hypertension, diabetes, ACD, male sex, obesity, renal failure (ESKD), and transplantation [8,9,10].

A study encompassing 21,817 ESKD patients from Taiwan reported a 64% increase in cancer risk for an ESKD patient compared to the general population [11]. Specifically looking at cancer of the kidney, an international cohort of 831,804 ESKD patients had a 3.60 times higher risk of renal malignancy than non-ESKD patients [2]. While clear cell renal cell carcinoma (ccRCC) is the most common histological subtype in both ESKD and non-ESKD patients, papillary RCC is reported to occur more frequently in the ESKD population [12,13,14]. Specific RCC subtypes have been linked to the length of time a patient received hemodialysis, with papillary RCC and acquired cystic disease-associated RCC (ACD-RCC) appearing more often in patients on dialysis for more than 10 years [15].

The association between ESKD and RCC may also be viewed from the opposite perspective. RCC has been reported to increase one’s risk of developing ESKD. An Australian study following 2739 RCC patients for 3 years examined how nephrectomy may cause a patient to need RRT [16]. Ellis et al. reported that 1 in 53 patients undergoing a radical nephrectomy due to RCC will develop novel ESKD within 3 years post-operation. While both radical and partial nephrectomy may lead to ESKD developing in the residual, non-neoplastic tissue, unsurprisingly, partial nephrectomy showed less correlation to loss of renal function, with only 1 in 100 patients developing ESKD.

Along with ESKD, kidney transplantation increases one’s risk of cancer, and, more specifically, increases one’s risk of RCC by 500% [17,18,19]. Figure 2 summarizes the various risk factors a KTR faces for developing RCC and groups them by their source. Kidney transplant recipients (KTRs) have their native kidney left in situ, which is significant, as roughly 90% of post-transplant RCC develops in the native or non-allograft kidney [20]. The increased RCC risk in transplantation patients is attributed to several factors, including acquired cystic disease, immunosuppressive drug regimens, and history of hemodialysis [21]. Some argue the increased risk is due to the close monitoring and follow-up provided to transplant recipients. KTRs are subjected to frequent radiographic imaging, symptom monitoring, and serum tests. Radiographic imaging has become widely available and more economic, leading to an increase in incidental identification of renal masses, serving as a potential explanation for the increased incidence of RCC in KTRs [21,22]. Data extracted from the United Network of Organ Sharing (UNOS) database reveal that the incidence of both pre- and post-transplant RCC has been steadily increasing since 1994, as seen in Figure 3 and Figure 4. As RCC is increasingly identified in ESKD patients and transplant donor and recipient pools continue to expand, it is vital for clinicians to better understand the triangular cause and effect relationship between RCC, ESKD, and transplantation.

## 3. Pathophysiology of Renal Malignancy in Patients with ESKD

The microenvironment of an ESKD patient serves as a unique physiologic entity compared to that of an ESKD patient post-renal transplantation. ESKD patients often present as terminally ill patients with a multitude of carcinogenic risk factors, making it difficult to uncover the exact etiology of RCC in this patient population. Oxidative stress is proposed as a potential mechanism explaining the higher incidence of RCC in ESKD patients compared to patients with normal renal function. Oxidative stress markers—iNOS, COX-2, and 8-OHdG—have been reported to be overexpressed in patients receiving long-term dialysis [23]. Oxidative stress in ESKD patients may directly influence hypermethylation of genetic programs leading to specific types of RCC. Oxidative damage has been associated with papillary RCC subtypes, although some argue that the subtype classification used to make this determination is outdated [15,23]. It is widely known that ESKD-related RCC is linked to an increased prevalence of the papillary subtype. Some argue that papillary RCC in ESKD patients displays distinct histologic and immunophenotypic features compared to papillary RCC in non-ESKD patients [24,25]. Both the increased prevalence and novel subtype support the notion that ESKD-related RCC follows a separate pathologic mechanism compared to RCC in the general population. Damaged tissue and reduced renal function present in an ESKD kidney may allow for opportunistic tumor growth that would be less likely in a non-ESKD patient due to an impairment in immunity [17].

ACD is a more widely accepted cause of RCC in ESKD patients. In 2016, the World Health Organization (WHO) officially declared ACD-associated RCC (ACD-RCC) a subtype of RCC [26]. ACD occurs in patients with chronic kidney disease (CKD) and is considered to be a consequence of sustained uremia [27]. Torres et al. report that roughly 90% of patients on dialysis for at least 8 years will develop ACD [28]. While ccRCC is the most common histological subtype in ESKD patients with less than 10 years of dialysis, ACD-RCC is associated with patients receiving dialysis for longer than 10 years [29]. ACD is characterized by the formation of cysts in dilated renal tubules. Cyst formation in ACD patients is attributed to tissue loss due to underlying renal insufficiency, leading to hyperplasia of epithelial cells [30,31]. Unregulated cell growth and proliferation in hyperplastic cystic tissue are proposed to be the cause of tumor growth and subsequent RCC [32]. Ishikawa et al. suggest the time spent on dialysis to be the most significant risk factor for developing ACD [33,34].

Beyond oxidative stress and ACD, hypoxic tissue damage in chronically diseased kidneys is debated to be related to pathologies seen in RCC. The literature suggests a connection between hypoxia and disruption of gene regulation pathways [35]. Despite receiving a substantial portion of the cardiac output, the kidneys function at variable levels of pO2, often referred to as hypoxic organs [36]. The poor oxygen delivery to renal tissue is attributed to the organ’s unique vasculature design, which favors secretory function while reducing blood flow to the capillary network [37,38,39].

Hypoxia-inducible factor prolyl hydroxylases (HIF-PHDs) are the main cellular oxygen sensors that regulate hypoxia-inducible transcription factors (HIFs). HIFs initiate gene expression in tissue experiencing hypoxia to restore appropriate oxygen levels. However, HIF not only regulates the cellular response to low oxygen levels but also controls the expression of over 500 genes, associated processes such as the cell cycle, angiogenesis, and cell growth [40]. Under normoxic conditions, excess HIF proteins are inactivated by the von Hippel-Lindau (VHL) ubiquitin ligase complex. As such, a reduction in VHL activity allows for constitutive activity of HIF, leading to inflammatory damage and unregulated cell growth.

RCC is characterized by inactivating VHL mutations, which may be germline and heritable or de novo and somatic. The dynamic relationship between HIF and VHL can illuminate the subtype, severity, and outcome of RCC. For instance, HIF variants such as HIF-1⍺ and HIF-2⍺ are associated with ccRCC, one of the most aggressive and common types of RCC seen in ESKD patients [29,41].

Hypoxia’s relationship with malignancy becomes more convoluted when considering the environment of an ESKD kidney. Hypoxic states lead to kidney injury by depriving renal cells of oxygen, causing fibrosis and irreversible damage to nephrons and capillary networks. While the HIF-VHL pathway is well understood regarding general RCC, the role HIF-VHL plays in a chronically diseased kidney is largely unclear [35]. Some argue that HIF promotes regrowth of damaged renal cells and controls inflammation, which are characteristic pathologies seen in ESKD kidneys [42]. Conversely, Faivre et al. postulate that HIF contributes to the increased cell damage and fibrosis seen in patients with CKD, citing the protective effects of HIF inhibitors [43]. Pinto et al. suggest this discrepancy highlights the notion that post-transcriptional modification of HIF proteins determines their pathological effect rather than solely its expression [44]. While the manner in which hypoxic environments contribute to the progression of ESKD is still being studied, it is widely accepted that hypoxia causing inflammation and fibrosis increases the risk of developing RCC, highlighting our current understanding of malignant pathophysiology in patients with ESKD.

It is important to note that, physiologically, there is no evidence to suggest that dialysis itself causes an increased carcinogenic risk. Dialysis prolongs a patient’s pre-transplant time. As dialysis does not alter renal pathology or fully replace renal function, it also prolongs the time a patient remains in renal failure and is exposed to the carcinogenic risks associated with ESKD. As outlined above, these carcinogenic risks induce a unique environment in the renal tissue that is susceptible to malignant growth. Tissue-specific alterations are the main perpetrators of the carcinogenic proliferation seen in ESKD patients and most likely continue to mold the microenvironment even after transplantation [45].

## 4. Pathophysiology of Renal Malignancy in Transplant Patients

As KTRs have been exposed to all the same risk factors associated with ESKD, the pathophysiology of cancer in transplant patients follows a similar pattern to that of an ESKD patient without a transplant. The main factors differentiating KTRs from ESKD patients are the use of immunosuppression therapy to mediate allograft rejection and the introduction of foreign renal tissue from the donor. The addition of immunosuppression and the donor kidney further alters the microenvironment of an ESKD patient. Immunosuppressive medication is hypothesized to allow for opportunistic oncogenic viruses to increase one’s risk of RCC. Historically, KTRs have been viewed through a similar lens to patients with human immunodeficiency virus (HIV). This perspective is supported by consistent evidence showing an increased risk of RCC in HIV/AIDS patients compared to the general population [46,47,48]. A smaller transatlantic case series refutes this notion by claiming the clinical presentation of RCC in HIV-positive and HIV-negative patients is similar, going on to suggest that immunosuppression plays a lesser role than traditional RCC risk factors such as age [49]. Their conclusion proposes that a suppressed immune system may not be the direct cause of de novo RCC in KTRs.

Karami et al. evaluated 683 KTRs with RCC and reported an elevated risk for ccRCC when polyclonal antibodies were used for induction therapy. The study also demonstrated an association between papillary RCC and IL-2 antagonists but was unable to provide a pathological explanation for either of these trends [14]. Calcineurin inhibitors (CNIs) are more widely known to contribute to renal malignancy. CNIs are toxic, reducing glomerular filtration and promoting fibrosis and tubular atrophy in the kidney [50,51]. Additionally, CNIs may contribute to tumor growth by their inhibition of DNA repair and apoptosis pathways [52]. As such, it is highly debated whether CNI’s immunosuppressive benefits outweigh its potential to induce neoplastic cellular changes. Hui et al. argue the combination of Belatacept (CD80/86-CD28 co-stimulation blocker) and mTOR inhibitors produce a similar immunosuppressive effect as CNIs, while avoiding the increased risk of cancer [53].

Interestingly, patients who have diabetes or polycystic kidney disease (PKD) as their indication for renal transplant are less likely to develop RCC compared to patients with vascular diseases, glomerular diseases, and hypertension [14,54,55]. PKD’s inverse relation to post-transplant malignancy has been hypothesized to be related to protective germline mutations and increased use of nephrectomy in PKD patients [55,56].

While both ESKD patients and post-transplantation patients share a higher risk of developing RCC, the exact physiological mechanism by which they differ is still being studied. It is of particular importance for transplant clinicians to better characterize how immunosuppressive medication type and dosage impact RCC development and graft outcomes.

## 5. RCC Outcomes in ESKD and Transplantation

As the list of ESKD patients waiting for a renal transplant continues to grow, more and more donors and recipients with a history of RCC are being considered for transplantation. According to the United Network of Organ Sharing (UNOS) database, malignancy is the third leading cause of death for kidney transplantation recipients (KTRs), behind cardiovascular events and infections, as seen in Figure 5. As such, our current understanding of how RCC develops in a KTR must be consolidated to include both donor and recipient factors.

### 5.1. Donor RCC

Excess ESKD and the shortage of organ donors have led to the use of expanded criteria for kidney donors, including the use of donors with a history of known renal malignancy. As such, ex vivo resection of renal masses from donor kidneys prior to transplant is not unprecedented [57,58]. Transplant of kidneys from both living and deceased donors with renal masses has been documented, with transplant outcomes appearing more favorable from living donor RCC [59,60].

Donors with a prior history of RCC are reported to have a higher risk of transmission compared to donors with breast and colon cancers [61]. He et al.’s prospective study followed 28 KTRs whose donors received a radical nephrectomy to remove small tumors (≤3 cm) prior to transplant. Only 7% of KTRs experienced allograft rejection and a median of 7.5 years of follow-up revealed no tumor reoccurrence [62]. The study supports the notion that the risk of transmitting RCC to the recipient is minimal given pre-transplant resection. A similar study that enrolled 10 living kidney donors undergoing nephrectomy for RCC was performed and found no tumor reoccurrence at 32 to 58 months post-transplant [63]. The study only included donors who had tumors smaller than 4 cm. While there was no tumor reoccurrence, 8 of the 10 transplant recipients experienced acute rejection, potentially due to the recipients being “high-risk”. The study supports the notion that donor RCC is unlikely to reoccur in the recipient but questions the impact that donor RCC and nephrectomy may have on graft survival. It is significant to note that all kidneys included in both studies were from living donors and tumor resection was performed just prior to the time of transplant. Only after the pathologist confirmed negative margins on the resected tumor did transplantation proceed.

That said, transmission of donor RCC to the renal transplant recipient is possible. UNOS has recorded a total of 33 cases of donor-transmitted RCC, although no study following these cases has been reported in the literature.

The current consensus surrounding the use of RCC kidneys in transplantation follows that ESKD patients who are transplanted with an RCC kidney maintain a significantly higher survival rate compared to those who would refuse transplantation in the hopes of receiving a healthier kidney and remain on the waitlist with dialysis [21,64].

### 5.2. Recipient RCC

ESKD patients are evaluated prior to being placed on the waitlist for a kidney. The evaluation includes a history and screening for malignancy, as this may affect their viability as a transplant recipient. Unfortunately, the shortage of donor organs forces transplant programs to prioritize ESKD patients with greater life expectancies. A patient with a history of invasive RCC will most likely have a shorter life expectancy post-transplant and, thus, is required to observe a period in between complete remission and transplantation. Currently, patients with a history of complex RCC are recommended to endure an observation period of 2 to 5 years depending on the extent of malignancy [65]. Many argue the observation period is too extensive and may be detrimental to ESKD patients in need of a transplant. The observation period prolongs an ESKD patient’s time on dialysis, which is correlated to worse transplant outcomes. Several studies have demonstrated the insignificance of observation periods for renal transplant patients as they are not correlated to tumor reoccurrence or recipient survival [66,67].

Compared to a general RCC patient, ESKD-related RCC is favorably associated with a younger age at diagnosis, smaller tumor size, more asymptomatic presentation, lower stage, and less metastases [68,69]. However, how oncologic outcomes differ between ESKD and non-ESKD patients remains inconsistent within the literature. Neuzillet et al. report that both ESKD-ccRCC and ESKD-papillary RCC show higher survival rates compared to their non-ESKD counterparts [68]. Hayami et al. refute this conclusion by reporting a significantly lower survival rate associated with RCC for dialysis patients compared to non-dialysis patients [69]. This notion is further supported when specifically examining RCC in long-term dialysis patients, who were found to have an increased mortality risk, even without an accompanying diagnosis of acquired cystic kidney disease [10].

A multicenter retrospective study out of France examined the role that renal transplant plays in RCC patient survival [70]. RCC arising in the native kidneys of transplant patients demonstrated favorable outcomes compared to ESKD patients who remained on dialysis without transplant and developed RCC. Transplant recipients had a significantly favorable 5-year survival rate at 97% compared to non-transplant patients with a 77% 5-year survival rate. Additionally, the renal tumors found in transplant recipients were associated with favorable staging and grading and a younger age of diagnosis, compared to their non-transplant ESKD counterparts. The advantageous findings related to RCC in transplant recipients are most likely due to the precocity of diagnosis; however, some suggest that the tumorigenesis process may differ between transplant and non-transplant RCC [70,71]. The improved survival could be explained by the more frequent follow-up and imaging given to post-operative transplant recipients compared to the routine oncologic screening seen in non-transplant-associated RCC.

A retrospective European study compared pre-transplant native-kidney RCC and post-transplant native-kidney RCC and found the time of RCC diagnosis relative to transplantation affected survival outcomes in KTRs [72]. The ten-year cancer-specific survival rates were 95% and 87% in the pre-transplantation and post-transplantation cohorts, respectively. The significance of malignancy in the recipient is demonstrated well in Van de Wetering’s nested case study reporting on 12,805 KTRs, where cancer caused 56% of all deaths when controlled for graft loss [73].

### 5.3. The Role of Immunosuppression

Solid organ transplantation is a well-known risk factor for cancer. Post-transplantation malignancy can be directly correlated to the use of long-term immunosuppression. Immunosuppression exposes patients to downregulated defense mechanisms and subsequently unregulated oncogenic viruses due to impaired immune surveillance in the host tissue [74].

Multiorgan transplants have been known to increase one’s risk of malignancy as they require extended immunosuppression regimens. Similarly, heart and lung transplantations are associated with a higher risk of cancer compared to renal transplantation as they require stronger regimens of immunosuppression [75]. Additionally, the type of immunosuppressive agent used may influence the magnitude of added risk. Calcineurin inhibitors (CNIs) have long been known to be associated with an increased risk of de novo cancers due to their underlying biologic mechanism and nephrotoxic profile. The harmful adverse effects associated with CNIs have led researchers to investigate alternative approaches to immunosuppression for renal transplantation.

The TRANSFORM study looked at incorporating everolimus, an mTOR inhibitor, into the immunosuppression regimen for KTRs, allowing for reduced CNI exposure [76]. The everolimus and reduced-CNI cohort displayed comparable adverse events compared to the control group, receiving a normal dosage of CNIs. Although their study population was not large enough to evaluate rare events such as RCC, the authors did cite a potential benefit of using mTOR inhibitors in routine immunosuppression protocols: minimization of the risk of post-transplant malignancy. Studies have demonstrated how an mTOR inhibitor may contribute to the secondary prevention of post-transplant neoplasms, specifically skin cancers [77,78]. The CONCEPT study demonstrated the benefit of Sirolimus (mTOR inhibitor) initiated in KTRs three months after transplant [79]. Although not the study’s primary endpoint, KTRs switched to Sirolimus versus KTRs remaining on Cyclosporine (CNI) demonstrated a lower incidence of post-transplant malignancy. However, to our knowledge, no study has questioned the potential benefit of mTOR inhibitor’s cytostatic effect on RCC risk post-transplant.

In our current understanding, there is no consensus on how immunosuppression regimens should be modified for patients that develop RCC after kidney transplantation. After following 17 solid organ transplant patients with RCC, Tollefson et al. concluded that patients with well-controlled, localized disease required no change in immunosuppressive management [80].

Despite the clear correlation between an immunosuppressed state and RCC, many argue that the increased incidence of RCC in transplant patients is due to underlying primary renal disease, citing mechanistic processes such as hemodialysis as the main perpetrator of malignancy rather than immunosuppression. Several studies refute this belief as the risk of malignancy after kidney transplantation retrogresses to pre-transplant levels after graft failure and discontinued immunosuppression [9,81,82]. With this said, it is widely accepted that the elevated risk RCC due to immunosuppression is outweighed by the mortality associated with chronic dialysis in ESKD without kidney transplantation. The decision to stop or alter immunosuppression must be personalized for each patient to balance the risk of malignancy and the risk of graft loss.

## 6. Treatment Modalities for RCC in ESKD Patients

When detected early, RCC generally displays favorable outcomes. Surgical resection of the tumor remains the first-line option for management of localized RCC (stages I–III) and is associated with a good prognosis. Partial nephrectomy is favored over radical nephrectomy to preserve renal tissue and maintain renal function. Radical nephrectomy has been linked to an increased risk of developing ESKD [16], again signifying the bidirectional relationship between RCC and ESKD. Advanced or complex RCC carries a worse prognosis, with roughly one third of patients progressing to metastatic disease [44].

Prior to the early 2000s, cytokine therapies, mainly interleukin-2 (IL-2) and interferon alfa (IFN-α), were the mainstay treatment option for patients with advanced RCC [83]. Developments in genomic research led to a further understanding of the molecular systems present in tumor cells, such as the vascular endothelial growth factor (VEGF), from which therapeutic agents like tyrosine kinase inhibitors (TKIs) were designed. Similarly, the mammalian target of the rapamycin (mTOR) pathway was identified to play a significant role in the unregulated cell proliferation seen in ccRCC, leading to the development and use of mTOR inhibitors. Targeted molecular therapies such as these have shown to be effective against advanced RCC, with similar results to the outdated cytokine-based agents but without the associated toxicity [84,85].

Spontaneous regression, T-cell infiltration, and response to immunotherapies support the classification of RCC as an immunologically active malignancy. As such, immune checkpoint inhibitors (ICIs) are now often considered first-line therapy, alongside molecular therapies like multi-TKIs [86]. Tumor cells associated with RCC display proteins on their cell membranes that hide their presence from T-cells attempting to identify foreign cells. ICIs have been developed to re-sensitize the cancerous cells to the host’s immune system, so that they may be labeled for cell death.

Unfortunately, in RCC patients who had undergone renal transplantation for ESKD, the use of ICIs has a negative impact on graft survival by increasing the potential for rejection. Although ICIs represent a serious risk for KTRs, other systemic therapies, such as anti-angiogenic drugs, hormone therapy, and platinum salts, are widely considered to be safer options for KTRs, although potentially less effective [87]. Providers should be mindful when administering anti-angiogenic drugs to treat RCC in ESKD patients as proteinuria, nephrotic syndrome, and diarrhea, all of which are common adverse effects, may worsen their already impaired renal function [88,89,90]. Unfortunately, data related to the impact of ICIs on RCC in transplant patients are scarce as these patients are often excluded from prospective trials.

The addition of mTOR inhibitors to help negate cancer progression in KTRs may be promising in terms of overall patient survival, although specific outcomes for complex RCC are still unknown [91,92,93]. Use of mTOR inhibitors in KTRs must be carefully monitored as they have been linked to impaired wound healing and thus should not be administered prior to oncologic surgical intervention for RCC, or prior to transplantation itself [87,94]. Nephron-sparing surgery remains a viable and effective treatment option for well-localized RCC in the graft kidney. Additionally, percutaneous-based focal treatment serves as an alternative management option for donor-transmitted RCC noted to be smaller than 4 cm. For ESKD patients, focal treatment may be preferred over traditional surgical management as it has been shown to have less impact on renal function [95,96].

## 7. Conclusions

RCC remains one of the deadliest complications after renal transplantation. RCC in the kidney transplant patient presents as a uniquely different histologic and pathologic entity than RCC in the ESKD patient. Clinicians treating and studying these patients must consider the impact of immunosuppression as well as the novel donor tissue present after transplantation. The current management strategy in this patient population is understudied. Several studies have examined the complex relationship between RCC, immunotherapy, and immunosuppression but no definitive protocol has surfaced. Oncologists addressing these patients must work to balance the tri-modal cause and effect relationship between RCC, ESKD, and transplantation.

## Figures and Tables

**Figure 1 cancers-16-00003-f001:**
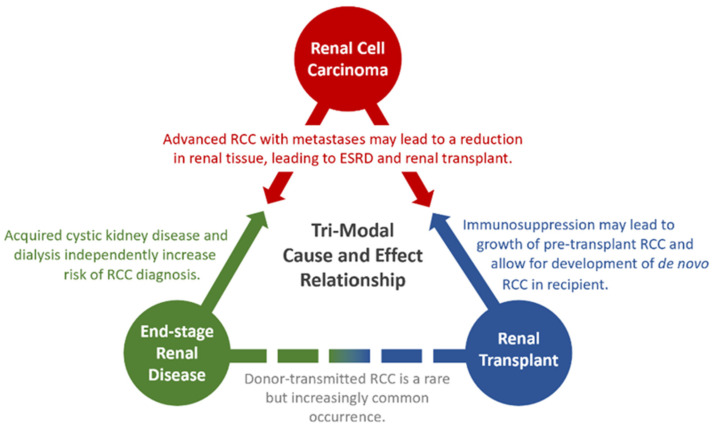
Triangular cause and effect diagram, visualizing the bidirectional relationships between RCC, ESKD, and transplantation.

**Figure 2 cancers-16-00003-f002:**
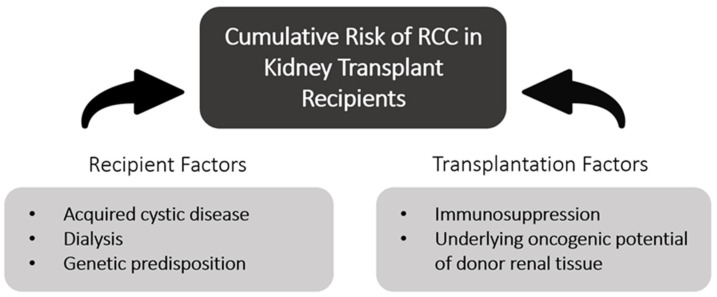
Diagram demonstrating the risks attributed to renal transplantation patients and their underlying origin (recipient-based or transplantation-based). All risk factors are discussed further in the review.

**Figure 3 cancers-16-00003-f003:**
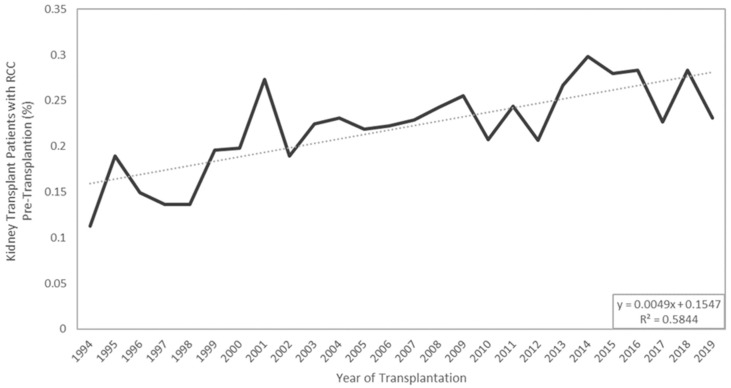
Longitudinal display of the incidence of kidney transplantation patients diagnosed with RCC at the time of transplantation. Data extracted from the UNOS database on 2 July 2022. Trendlines displayed.

**Figure 4 cancers-16-00003-f004:**
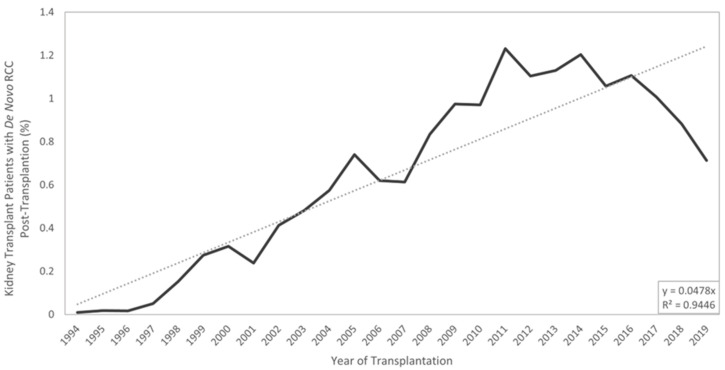
Longitudinal display of the incidence of kidney transplantation patients diagnosed with de novo RCC after transplantation. Data extracted from the UNOS database on 2 July 2022. Trend lines displayed.

**Figure 5 cancers-16-00003-f005:**
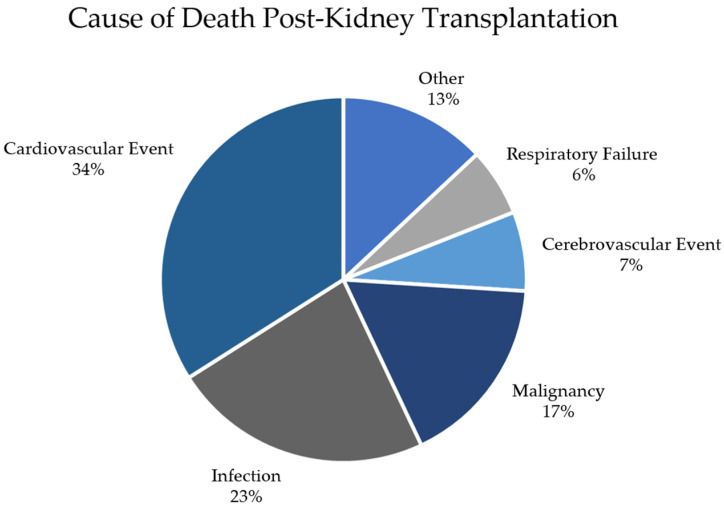
Pie chart displaying the leading causes of death for patients after receiving a kidney transplantation. Data were extracted from the UNOS database on 2 July 2022.

## Data Availability

Data are unavailable due to privacy and ethics considerations.

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
