# Peer review of "Renal Cell Carcinoma in End-Stage Kidney Disease and the Role of Transplantation"

_cancers, 2023, doi:10.3390/cancers16010003_

Round 1

Reviewer 1 Report

Comments and Suggestions for Authors

Thank you to Robinson et al for submitting this review manuscript exploring RCC (kidney cancer) amongst patients with ESKD (kidney failure), including kidney transplant. It is well structured and useful, with some very nicely delivered & explored insights. I have some queries:

1) Please consider changing ESRD to ESKD and RRT to KRT to align with current nomenclature. RCC should remain RCC, and KTR should remain KTR.

2) Figure 1 - strongly suggest to remove "but increasingly common occurrence" at the bottom of the figure in relation to donor-transmitted RCC. This statement is neither supported by your text nor the broader literature at this time.

3) Line 74 - Tuberous sclerosis is mentioned here. Can you give any additional reference or explanation here?

4) Line 98 - please change ACKD to ACD as previously used. Given that ACD is the acronym used throughout, I would suggest using it only unless there is a specific reason to change.

5) Figure 3, Figure 4 - please give the date of database access and the website

6) Line 136-137 - this doesn't entirely make sense. "would be inaccessible in a non-ESRD patient" is unclear. Is this referring to immune cells??

7) Line 143 - correct CKD to ACD

8) italicise gene names please

9) Line 168-169 - suggest to change to "which may be gremline and heritable or de novo and somatic". 

10) Figure 5 - please give the website access through

Author Response

Reviewer Comment

Author Comments and Revisions

Reviewer 1

Thank you to Robinson et al for submitting this review manuscript exploring RCC (kidney cancer) amongst patients with ESKD (kidney failure), including kidney transplant. It is well structured and useful, with some very nicely delivered & explored insights. I have some queries:

1) Please consider changing ESRD to ESKD and RRT to KRT to align with current nomenclature. RCC should remain RCC, and KTR should remain KTR.

All nomenclature changes were made, including within figures.

2) Figure 1 - strongly suggest to remove "but increasingly common occurrence" at the bottom of the figure in relation to donor-transmitted RCC. This statement is neither supported by your text nor the broader literature at this time.

The authors agree with your opinion. Figure 1 was edited to reflect this.

3) Line 74 - Tuberous sclerosis is mentioned here. Can you give any additional reference or explanation here?

The authors appreciate the reviewer’s insight on the inclusion of tuberous sclerosis. While associated with RCC, the authors agree that referring to tuberous sclerosis as a “risk factor” may not be proper terminology. As such, it was removed from the sentence. If the reviewer is interested in TB and RCC a strong piece of literature on the two conditions can be found at the following reference. https://doi.org/10.1038/sj.ki.5001853

4) Line 98 - please change ACKD to ACD as previously used. Given that ACD is the acronym used throughout, I would suggest using it only unless there is a specific reason to change.

 The authors agree and have adjusted the wording and acronym.

5) Figure 3, Figure 4 - please give the date of database access and the website

Each figure legend was edited to include the data of access; however, a specific website cannot be provided as the data was pulled from a subscription-based report from UNOS. A website address may be added leading to the UNOS’s data available for free however this would not contain the data shown in the figure.

6) Line 136-137 - this doesn't entirely make sense. "would be inaccessible in a non-ESRD patient" is unclear. Is this referring to immune cells??

The sentence was edited and a new citation was added to provide a better explanation of this concept.

7) Line 143 - correct CKD to ACD

Change made.

8) italicise gene names please

Von Hippel-Lindau and hypoxia-inducible factor were italicized within the text.

9) Line 168-169 - suggest to change to "which may be gremline and heritable or de novo and somatic". 

 The authors appreciate the suggest and have made the change.

10) Figure 5 - please give the website access through

A specific website cannot be provided as the data was pulled from a subscription-based report from UNOS. A website address may be added leading to the UNOS’s data available for free, however this would not contain the data shown in the figure.

Reviewer 2 Report

Comments and Suggestions for Authors

In a manuscript entitled "Renal Cell Carcinoma in End-Stage Renal Disease and the Role of Transplantation," SG Robinson and coworkers review the literature on renal cell carcinoma in the context of end-stage kidney disease in patients maintained on dialysis and renal cell carcinoma in renal transplant recipients. 

The review is exhaustive and provides a wealth of information, including updated and most recent referenced sources.

A major limitation of this review that it is responsible for considerable confusion is a failure to recognize different contexts from which renal cell carcinoma is identified.  The renal cell carcinoma in each of these contexts has a different pathogenesis, morphology and clinical as well as laboratory features.  Thus, in the subsequent sections one gets confused on what type of renal cell carcinoma in question.

In principle, the following types of renal cell carcinoma are recognized in the context of this review   These points should be clarified from the outset.

1. Renal cell carcinoma developed in end-stage renal disease patients who are maintained on long-term dialysis.

2. Renal cell carcinoma developed in native kidneys of patients who received renal transplantation.

3. Renal cell carcinoma developed in transplanted kidney of the renal transplant recipients.

4. Renal cell carcinoma in donors with a clinical history of renal cell carcinoma.

5. Renal mass/ renal cell carcinoma identified in donated organs at the time of donor harvesting.

In addition, the authors also put a lot of emphasis upon a condition in which renal cell carcinoma in the general population, treated with a partial or total nephrectomy leading to end-stage kidney disease. Although important, this topic is quite different from the entities covered in this review and is irrelevant to this review. Adding this is irrelevant and creates a lot of confusion.

 To alleviate this problem, the authors may want to clarify the matter from the beginning and in subsequent sections clarify the type of renal cell carcinoma in focus.

The following are some minor points calling for modification.

Lines 16-25 (abstract): This should be rewritten to clarify the matter detailed above.

Lines 34-35:  This statement may not be accurate.  Please also refer to Lines 298 and 300 for a contrasting and probably corrected view.

Lines 41 and 44:  As mentioned above, this paragraph refers to a completely different matter, namely, renal cell carcinoma in the general population, leading, not by itself, but by partial or total nephrectomy, to end-stage kidney disease. 

Lines 85-91:  This gives the reader the feeling that renal cell carcinoma leading to ESRD is not accurate.  Renal cell carcinoma by itself usually does not cause end-stage kidney disease.  Only after partial or total nephrectomy for renal cell carcinoma, then the end-stage disease develops in the residual, non-neoplastic kidney tissue.

Lines 85-90:  As mentioned above, this creates confusion because it refers to the matter of end-stage disease developing after renal cell carcinoma in the general population being treated with partial or total nephrectomy. 

Line 143:  The letters CKD should be changed to ACK. 

In Section 4, discussing the pathogenesis of renal cell carcinoma, the authors may want to pay attention to the following point.  The authors may want to briefly mention the following points:  a) renal cell carcinoma in native kidney with end-stage renal disease vs. renal cell carcinoma in native kidney in renal transplant patients vs. renal cell carcinoma in transplanted kidney; b) the histologic types of renal cell carcinoma in native kidney with end-stage renal disease vs. histologic type of renal cell carcinoma in kidney still with end-stage renal disease but now receiving renal transplant. The authors then may want to key in these points in discussing the pathogenesis and clinical behaviors of the different forms of renal cell carcinoma.

Section 5.1:  This review may create an impression that it is quite acceptable to use kidney in patients with renal cell carcinoma for transplant.  The authors may want to emphasize the stage, the grade, the histologic type, the presence or absence of tumor at the time of organ donation, and the duration of remission in this context. These findings may have an impact on the donor selection.  Furthermore, the authors may want to emphasize different concepts between donors with a history of renal cell carcinoma vs. a donor with a renal mass/renal cell carcinoma identified only at organ harvesting. 

In Section 6, Lines 362-384, these discussions are focused on the management of renal cell carcinoma in the generation population, emphasizing the underlying principle of control tumor at the same time preserving as much nephron mass as possible.  These considerations are probably irrelevant to the current discussion in which renal cell carcinoma develops in the background of end-stage kidney disease. Whether these considerations are applicable to the renal cell carcinoma developing de novo in the transplanted kidney is another matter.  The authors may want to refer to this also.

Comments on the Quality of English Language

See above

Author Response

Reviewer 2

A major limitation of this review that it is responsible for considerable confusion is a failure to recognize different contexts from which renal cell carcinoma is identified.  The renal cell carcinoma in each of these contexts has a different pathogenesis, morphology and clinical as well as laboratory features.  Thus, in the subsequent sections one gets confused on what type of renal cell carcinoma in question.

In principle, the following types of renal cell carcinoma are recognized in the context of this review   These points should be clarified from the outset.

1. Renal cell carcinoma developed in end-stage renal disease patients who are maintained on long-term dialysis.

2. Renal cell carcinoma developed in native kidneys of patients who received renal transplantation.

3. Renal cell carcinoma developed in transplanted kidney of the renal transplant recipients.

4. Renal cell carcinoma in donors with a clinical history of renal cell carcinoma.

5. Renal mass/ renal cell carcinoma identified in donated organs at the time of donor harvesting.

The authors appreciate Reviewer 2’s careful reading and thoughtful commentary on the manuscript. However, the authors politely disagree with this particular comment. The complex cause-and-effect relationship between RCC, ESKD, and kidney transplantation is what warrants the review. The unique characteristics of RCC within different renal tissue contexts, as mentioned in the reviewer comment, are thoughtfully discussed in different sections of the manuscript. With the exception of type number 4, each RCC type is mentioned in the review and the authors feel each circumstance is clarified accurately. If specific lines or paragraphs of the text are leading to confusion the authors should and will address those areas. RCC type 4 is not mentioned in the review as the literature search revealed little to no significance for RCC in the recipient when no active or recent history of RCC in the donor was present at the time of organ procurement. Again, the authors are grateful for all of Reviewer 2’s suggestions as they have led to a careful analysis of the writing. No changes were made in response to this particular comment.

In addition, the authors also put a lot of emphasis upon a condition in which renal cell carcinoma in the general population, treated with a partial or total nephrectomy leading to end-stage kidney disease. Although important, this topic is quite different from the entities covered in this review and is irrelevant to this review. Adding this is irrelevant and creates a lot of confusion.

The authors feel that the discussion of nephrectomy for RCC in the general population is important to include as it helps explain the complex nature of RCC and ESKD. How, within select circumstance RCC may increase risk for ESKD and vis-vera, ESKD may increase risk for RCC.

Lines 16-25 (abstract): This should be rewritten to clarify the matter detailed above.

The authors have made no change to the abstract. Please see the response to Reviewer 2’s first comment.

Lines 34-35:  This statement may not be accurate.  Please also refer to Lines 298 and 300 for a contrasting and probably corrected view.

The authors appreciate this point and have edited the statement made on lines 34-35 to reflect this.

Lines 41 and 44:  As mentioned above, this paragraph refers to a completely different matter, namely, renal cell carcinoma in the general population, leading, not by itself, but by partial or total nephrectomy, to end-stage kidney disease. 

The authors feel that the sentence on lines 43-44 reflect this notion, alluding to the fact that RCC leads to ESRD by the treatment modalities including nephrectomy, not solely RCC alone. “Treatment modalities for RCC, including both nephrectomy and anti-cancer medication, reduce functional renal tissue, thus increasing one’s risk of CKD and ESKD.”

Lines 85-91:  This gives the reader the feeling that renal cell carcinoma leading to ESRD is not accurate.  Renal cell carcinoma by itself usually does not cause end-stage kidney disease.  Only after partial or total nephrectomy for renal cell carcinoma, then the end-stage disease develops in the residual, non-neoplastic kidney tissue.

The authors agree that this sentence gave a wrongful impression and have edited the lines to better explain the concept. (lines 84-86)

Lines 85-90:  As mentioned above, this creates confusion because it refers to the matter of end-stage disease developing after renal cell carcinoma in the general population being treated with partial or total nephrectomy. 

The authors agree that this sentence gave a wrongful impression and have edited the lines to better explain the concept. (lines 84-86)

Line 143:  The letters CKD should be changed to ACK. 

The authors recognize the limitation of the cystic kidney disease (CKD) acronym and have changed all in-text mentions of CKD to acquired cystic disease (ACD).

In Section 4, discussing the pathogenesis of renal cell carcinoma, the authors may want to pay attention to the following point.  The authors may want to briefly mention the following points:  a) renal cell carcinoma in native kidney with end-stage renal disease vs. renal cell carcinoma in native kidney in renal transplant patients vs. renal cell carcinoma in transplanted kidney; b) the histologic types of renal cell carcinoma in native kidney with end-stage renal disease vs. histologic type of renal cell carcinoma in kidney still with end-stage renal disease but now receiving renal transplant. The authors then may want to key in these points in discussing the pathogenesis and clinical behaviors of the different forms of renal cell carcinoma.

The authors feel the part a) of the reviewer’s comments were addressed properly within sections 3 and 4 of the manuscript. Regarding part b) of the reviewer’s comments, the authors agree that the histological subtypes of RCC play a role in the pathogenesis of the malignancy; however, the histologic subtype between the two patient populations mentioned do not differ greatly and including the information within the context of this review article would be outside the intended scope, additionally making the review unnecessarily long. This information would be valuable, however most likely better read in a separate article.

Section 5.1:  This review may create an impression that it is quite acceptable to use kidney in patients with renal cell carcinoma for transplant.  The authors may want to emphasize the stage, the grade, the histologic type, the presence or absence of tumor at the time of organ donation, and the duration of remission in this context. These findings may have an impact on the donor selection.  Furthermore, the authors may want to emphasize different concepts between donors with a history of renal cell carcinoma vs. a donor with a renal mass/renal cell carcinoma identified only at organ harvesting. 

Section 5.1 was edited to reflect this distinction between donors with a history of renal cell carcinoma vs. a donor with a renal mass/renal cell carcinoma identified only at organ harvesting. Additionally, when data was available information on the tumor stage/grade was included for the mentioned studies.

In Section 6, Lines 362-384, these discussions are focused on the management of renal cell carcinoma in the generation population, emphasizing the underlying principle of control tumor at the same time preserving as much nephron mass as possible.  These considerations are probably irrelevant to the current discussion in which renal cell carcinoma develops in the background of end-stage kidney disease. Whether these considerations are applicable to the renal cell carcinoma developing de novo in the transplanted kidney is another matter.  The authors may want to refer to this also.

The authors appreciate the reviewers point on the relevance of information revolving around the management of renal cell carcinoma in the general population but did not make any changes to the text as they feel this information may provide context into how RCC may be managed in a way that increases risk for chronic kidney disease to develop, via nephrectomy.

Reviewer 3 Report

Comments and Suggestions for Authors

The manuscript "Renal Cell Carcinoma in End-Stage Renal Disease and the Role of Transplantation" by Samuel G Robinson and co-workers is a literature review on the important topic of risk from renal cell carcinoma in the kidney-transplanted patients. The manuscript has been carefully prepared and provides a comprehensive overview of the aforementioned field. The article contains five attractive and carefully prepared corresponding figures. I cannot find any serious deficiencies. Minor suggestion would be to show the data from figure 5 rather as a pie-chart. I therefore suggest that you publish the article after a minor revision.

Author Response

Reviewer Comment

Author Comments and Revisions

Reviewer 3

The manuscript "Renal Cell Carcinoma in End-Stage Renal Disease and the Role of Transplantation" by Samuel G Robinson and co-workers is a literature review on the important topic of risk from renal cell carcinoma in the kidney-transplanted patients. The manuscript has been carefully prepared and provides a comprehensive overview of the aforementioned field. The article contains five attractive and carefully prepared corresponding figures. I cannot find any serious deficiencies. Minor suggestion would be to show the data from figure 5 rather as a pie-chart. I therefore suggest that you publish the article after a minor revision.

The authors appreciate reviewer 3’s comments and agree with their suggestion regarding figure 5. Figure 5 has been altered to appear as a pie chart.